# Does Training Improve Sanitary Inspection Answer Agreement between Inspectors? Quantitative Evidence from the Mukono District, Uganda

**Richard King** [1,*], **Kenan Okurut** [2], **Jo Herschan** [1], **Dan J. Lapworth** [3], **Rosalind Malcolm** [4], **Rory Moses McKeown** [5] **and Katherine Pond** [1]

1   Department of Civil and Environmental Engineering, University of Surrey, Guildford, Surrey GU2 5XH, UK; j.herschan@surrey.ac.uk (J.H.); k.pond@surrey.ac.uk (K.P.)
2   Department of Civil and Building Engineering, University of Kyambogo, Kiwatule—Banda, Kampala, Uganda; ken_okurut@yahoo.com
3   British Geological Survey, Maclean Building, Wallingford OX10 8BB, UK; djla@bgs.ac.uk
4   School of Law, University of Surrey, Guildford, Surrey GU2 5XH, UK; r.malcolm@surrey.ac.uk
5   Consultant, World Health Organization, CH-1211 Geneva, Switzerland; RoryMosesMcKeown@live.com
*   Correspondence: r.a.king@surrey.ac.uk; Tel.: +353-87-615-5994

**Abstract:** Sanitary inspections (SIs) are checklists of questions used for achieving/maintaining the safety of drinking-water supplies by identifying observable actual and potential sources and pathways of contamination. Despite the widespread use of SIs, the effects of training on SI response are understudied. Thirty-six spring supplies were inspected on two occasions, pre- and post-training, by an instructor from the research team and four local inspectors in the Mukono District of Uganda. SI score agreement between the instructor and each inspector was calculated using Lin's concordance correlation coefficient. Average SI score agreement between the instructor and all inspectors increased post-training for the Yes/No answer type (0.262 to 0.490). For the risk level answer type (e.g., No, Low, Medium, High), average SI score agreement between the instructor and all inspectors increased post-training (0.301 to 0.380). Variability of SI scores between the four inspectors was calculated using coefficient of variation analysis. Average SI score variability between inspectors reduced post-training for both answer types, Yes/No (21.25 to 16.16) and risk level (24.12 to 19.62). Consistency of answer agreement between the four inspectors for each individual SI question was calculated using index of dispersion analysis. Average answer dispersion between inspectors reduced post-training for both answer types, Yes/No (0.41 to 0.27) and risk level (0.55 to 0.41). The findings indicate that training has a positive effect on improving answer agreement between inspectors. However, advanced training or tailoring of SI questions to the local context may be required where inconsistency of responses between inspectors persists, especially for the risk level answer type that requires increased use of inspector risk perception. Organisations should be aware of the potential inconsistency of results between inspectors so that this may be rectified with appropriate training and, where necessary, better SI design and customisation.

**Keywords:** drinking-water quality; microbial contamination; risk assessment; risk management; sanitary survey; training; water safety planning

## 1. Introduction

Until the early 2000s, investigation into sources of contamination following a noncompliant water quality test result had been the customary process for managing drinking-water safety. The reactive nature of this approach, coupled with the inherent challenges associated with water quality

testing [1], meant it was deemed insufficient as a standalone activity for ensuring the safe management of drinking-water supplies [2]. The World Health Organization (WHO) [3] recommends water quality testing and risk assessment be undertaken as complementary activities, where possible. To accurately estimate the safety of a supply, results from both approaches are required to confirm the absence of contamination and the low probability of future contamination [4].

One approach to the risk assessment of drinking-water supplies is the use of sanitary inspections (SIs). Defined by the WHO [5] as "an on-site inspection of a water supply to identify actual and potential sources of contamination", SIs provide a low-cost, easy-to-use monitoring approach that is particularly suited to small systems and settings with limited resources and/or capacity [6]. WHO SI forms focus on distinct point-source supply types (such as springs) and methods of water storage and distribution. Presented as a list of equally weighted "Yes/No" questions that indicate the presence or absence of observable contaminant pathways; actual and potential sources of contamination; and breakdowns in barriers to contamination [7], the output of the WHO SI forms is calculated by tallying the number of identified risk factors at a supply to provide a sanitary risk score. This risk score is then used to categorise the level of risk at the supply from "low" to "very high".

Sanitary risk scores and water quality test results do not exhibit a consistent positive linear relationship [8–17]. The dynamic nature of both risk and water quality, particularly that indicated by faecal indicator bacteria; the limitations associated with water quality testing; and the often non-standard, potentially subjective [18] SI approach means one metric cannot always be used to reliably predict or infer the other. Furthermore, whereas SI scores represent actual and potential risk to water quality, water test results only indicate the immediate quality of water sampled at a specific point in time. While such non-comparability questions the suitability of direct comparisons of some water quality parameters, such as faecal indictor bacteria, with SI risk scores, the consistency of SI answer agreement (or lack thereof) between inspectors still needs to be assessed while acknowledging the complexity of the SI process.

Risk perception, as defined by Darker and Whittaker [19], is a subjective judgment that people make about the characteristics and severity of a risk. The psychological parameters that affect risk perception [20–22], and more specifically, inter-inspector agreement [23], have been discussed by others. Studies of risk perception within industries such as aviation [24–26] have shown diverse trends in risk perception between experts, professionals and novices. Though the importance of both capacity building within the wider water sector [27] and, more specifically, suitable training prior to undertaking an SI is acknowledged in the literature [6,28], there is limited published data regarding the consistency of SI inter-inspector answer agreement [29,30]. However, such consistency is vital to allow for potential comparability of SI outputs.

The WHO SI forms are currently being updated to make them more robust; reflect appropriate technologies alongside current best practice technical and management advice; and to better align with the Water Safety Plan (WSP) methodology [6]. As the drinking-water sector continues to embrace the risk-based approach and expand the scope of mandatory risk assessment within legislation, for example, within the recent draft recast of the European Union (EU) Drinking-water Directive [31], consistent agreement of SI results between inspectors is crucial for the potential to reliably compare SI data within organisations and across regions. The potential for such consistency within an inspector group could be affected by a number of factors, such as SI form content and design; inspector experience and training; and weather conditions during the SI (if undertaken at different times) [6].

The objective of this study was to investigate the effect of training on inspector risk perception and inter-inspector answer agreement. Whilst Okotto-Okotto et al. and Yentumi et al. [29,30] varied study design aspects, such as inspector experience and inspection times, this was the first study to maintain a constant before/after standardisation of study design elements (e.g., water supply locations, inspectors, SI forms), except for controlled differences in inspector training between study 1 and 2. As described further in the Materials and Methods section, a modified SI form was developed and applied in the current study, where SI questions were designed using both the traditional "Yes/No" answer type and

the risk level assignment answer type (e.g., No, Low, Medium, High) that is commonly utilised in WSPs. The latter answer type was analysed due to the clear and increasing alignment between SIs and WSPs [6], however, unlike the prescriptive risk definitions often generated from WSP risk matrices, risk levels were used in this study to broadly define the overall risk to water quality. A second objective was to identify individual SI questions that caused the greatest inconsistency in answer agreement amongst inspectors for both answer types.

## 2. Materials and Methods

### 2.1. Study Area and Supply Type Selection

Spring supplies were chosen for inspection in this study due to their non-standard configurations compared to boreholes and other groundwater sources. Such variation in supply design presents a greater potential for differences in perception of risk among inspectors. Fieldwork was conducted in the Goma subcounty of Mukono District, a neighbouring district of the Ugandan capital, Kampala. Springs are the most common water source type in the Mukono district, and Goma subcounty has the most spring sources and the second lowest access to safe water in Mukono District [32]. The Mukono District Water Office helped to identify 76 spring sources in the Goma subcounty, 39 of which were functioning and in use. The Morgan formula [33] was used to calculate the representative sample size of 36 springs from the population of functioning springs. The locations and broad spatial coverage of the selected point-source springs throughout the subcounty are shown in Figure 1.

### 2.2. SI Form Development

During the study period, the *Guidelines for Drinking-water Quality, 2nd edition: Volume 3—Surveillance and control of community supplies* [34] were being updated to reflect the most current guidance and best practices relevant to the safe management of small drinking-water supplies. The University of Surrey, UK, supported the review and update of the SI forms to be included in the 3rd edition of the *Guidelines* [6]. As a result of this collaboration, a prototype spring SI form was used in this study (Table 1). The prototype form had been subject to updates from the original 1997 version, having undergone peer review by an expert group tasked with authoring the *Guidelines* revision, and a pilot review from 23 experts across 13 organisations. The prototype form consists of 12 revised questions with accompanying explanatory notes that were developed for small spring supplies based on an extensive literature review. Questions were not tailored specifically for the Ugandan context. Questions are answered "Yes" if a risk factor is present and "No" if a risk factor is not present, or not applicable. If a question is answered "Yes", indicating the presence of a risk factor, the inspector is instructed to complete a further step on the form and grade the risk as "Low" (1), "Medium" (3) or "High" (5). For example, three "High" and two "Medium" risks will provide a sanitary risk score of 21. The prototype form also includes an additional column to prompt inspectors to suggest remedial actions to identified risks.

**Table 1.** Modified sanitary inspection (SI) form used in study 1 and 2.

| | Sanitary Inspection Questions | NO | YES (Risk) | Risk Level (Circle Risk *Only* if YES Is Ticked) | | | What Action Is Needed? |
|---|---|---|---|---|---|---|---|
| 1 | Is the masonry, concrete wall or spring box absent or inadequate to prevent contamination? It is important that spring water is not exposed to contamination during the period of time between leaving the ground and being collected by the user. Masonry, a concrete wall or a spring box will protect the water from contamination during this time period. [a] | | | Low | Medium | High | |
| 2 | If there is a spring box, is the inspection cover or overflow pipe absent or inadequate to prevent contamination? A missing or inadequate (e.g., damaged, corroded, cracked, leaking) inspection cover or overflow pipe may increase the likelihood of contamination entering the spring box. If present, they should not allow the entry of vermin and other pollution into the spring box. | | | Low | Medium | High | |
| 3 | If there is a spring box, and there is an air vent, is it inadequately covered to prevent contamination? An air vent that is open to the environment may increase the likelihood of contamination entering the spring box. If present, the air vent should not allow the entry of vermin and other pollution into the spring box. | | | Low | Medium | High | |
| 4 | If there is a spring box, does it contain any visible sign of contamination (e.g., animal waste, sediment accumulation)? Contamination in the spring box may constitute a risk to water quality. Small deposits of silt at the bottom of the spring box are less likely to threaten water quality compared to animal waste, floating solids or biological growth. | | | Low | Medium | High | |
| 5 | Is the backfill area eroded or prone to erosion due to absence of vegetation? If the backfill area (directly behind the spring box or concrete wall) becomes eroded (e.g., due to absence of vegetation), it may act as a direct pathway for contamination to enter the spring water before it is collected by the user. | | | Low | Medium | High | |
| 6 | Is the fencing or barrier around the spring absent or inadequate to prevent contamination? If there is no fence or barrier around the spring (or if the fence is damaged or not fit for purpose), animals can access the spring site and may damage the structure as well as pollute the area with excreta. | | | Low | Medium | High | |
| 7 | Is the fencing or barrier upstream of the spring inadequate to stop local pollution? [b] If there is no fence/barrier upstream of the spring (or if the fence/barrier is damaged or not fit for purpose) then the shallower groundwater may become contaminated as it approaches the spring structure. | | | Low | Medium | High | |
| 8 | Is a storm water diversion ditch above the spring absent or inadequate to prevent contamination? If the diversion ditch is absent or inadequate (e.g., blocked, not wide or deep enough), contaminated surface water may enter the spring facility from above during periods of rain, or other events that may cause excess water to flow down towards the spring site. | | | Low | Medium | High | |
| 9 | Is there a latrine, septic tank or sewer line within 10 meters of the spring? Latrines close to groundwater supplies may affect water quality (e.g., by infiltration). You may need to visually check structures to see if they are latrines, in addition to asking residents about the presence of septic tanks and sewer lines. | | | Low | Medium | High | |

**Table 1.** *Cont.*

| | Sanitary Inspection Questions | NO | YES (Risk) | Risk Level (Circle Risk *Only* if YES Is Ticked) | | | What Action Is Needed? |
|---|---|---|---|---|---|---|---|
| 10 | Is there a latrine, septic tank or sewer line on higher ground within 30 meters of the spring? Pollution on higher ground poses a risk, especially in the wet season, as faecal material may flow into the spring. Groundwater may also flow towards the spring from the direction of the latrine, septic tank or sewer line. | | | Low | Medium | High | |
| 11 | Can signs of other sources of pollution be seen within 10 meters of the spring (e.g., animals, rubbish, human settlement, open defecation)? Animals or human faeces on the ground close to the spring constitute a serious risk to water quality. Presence of other waste (household, laundry, agricultural etc.) also constitutes a risk to water quality. | | | Low | Medium | High | |
| 12 | Is there an open/uncapped well or borehole within 100 meters of the spring? Any point of entry to the aquifer that is unprotected is a direct pathway for contaminants to enter the spring. | | | Low | Medium | High | |

| | | Risk level | Number of risks | Multiply by: | Score |
|---|---|---|---|---|---|
| a. | In some cases, masonry or a concrete wall may be in place instead of a spring box—provided they are in good condition and fit for purpose, this is acceptable from a water safety perspective. | Low | | 1 | |
| b. | Adequate fencing or barrier implies that the upstream area is closed off to where the groundwater is at least 2 meters deep or 30 meters away from the eye of the spring. | Medium | | 3 | |
| | *Note:* Enter the number of 'Low', 'Medium', 'High' risks and multiply by the relevant number to generate a 'Score'. The sum of the three scores is the 'Sanitary risk score'. | High | | 5 | |
| | | Sanitary risk score (max. 60) | | Total: | |

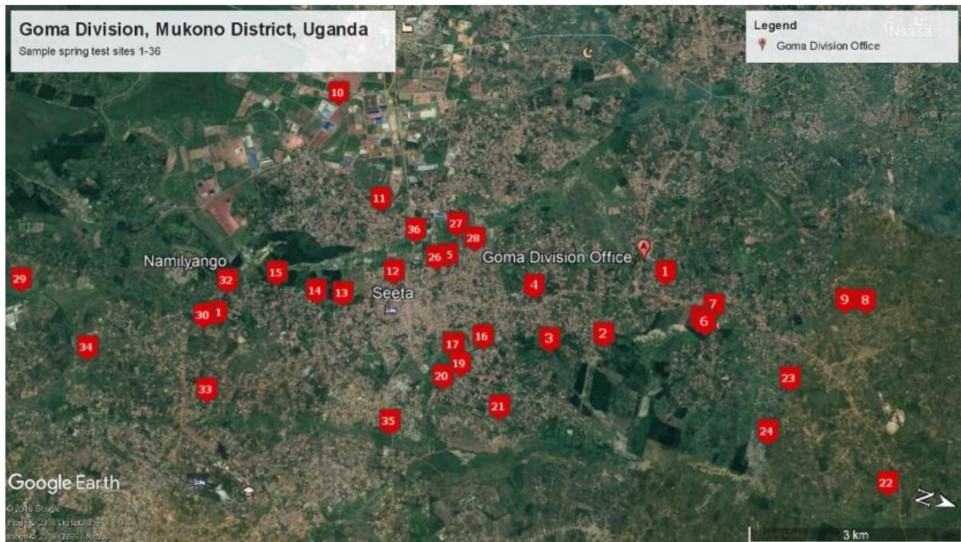

**Figure 1.** Locations of inspected spring supplies in Uganda (taken from Google Earth $^{\text{TM}}$, 2018).

*2.3. Fieldwork and Inspector Recruitment and Training*

Four inspectors and an instructor were used to undertake SIs in this study. The instructor was part of the University of Surrey research team and had significant background knowledge of the SI process and extensive experience working on the update of the entire WHO SI form suite. Inspectors 1–4 were all master's students studying water engineering at Kyambogo University, Uganda. Each had over five years' experience working in the drinking-water sector, though none had previously used an SI form. Fieldwork was undertaken over two study periods: study 1 (pre-training) was undertaken in April and study 2 (post-training) was undertaken in June/July. Prior to study 1, inspectors received a 45 min introduction to the concept of SIs. No specific details relating to SI form content were discussed, only the background to the approach. Prior to study 2, in addition to their previous introduction to SIs and the experience they had gained during study 1, inspectors 1–4 received a further four-hour training session at a spring supply where they were given the opportunity to ask specific questions relating to spring supply component names, scale of risk and relationships between risk factors.

*2.4. Study Design*

All of the inspectors travelled together to inspect each of the 36 supplies at the same time during both studies. The research team mitigated against the potential influencing of results by ensuring there was no inter-inspector discussion relating to SIs throughout the fieldwork. The research team mitigated against the Hawthorne effect—the alteration of behaviour by inspectors due to their awareness of being observed [35]—by emphasising the subjective nature of the exercise and by avoiding direct observation of inspectors during inspections. The potential effects of seasonal variability on study design were considered, however, the research team concluded that this should not affect the potential to gauge consistency of answer agreement if all inspectors undertook inspections at the same supply under the same conditions. Regardless, unpredictable rainfall patterns were observed by the research team throughout both studies, resulting in no differentiation between the two studies in terms of rainfall.

*2.5. Data Analysis*

2.5.1. Pearson's Correlation Coefficient and Lin's Concordance Correlation Coefficient (Lin's CCC)

Using Pearson's correlation coefficient, the correlation of SI scores between the instructor and each of the inspectors was determined at every supply for both answer type (Yes/No and risk level), pre- and post-training. Lin's CCC was used to determine the consistency of SI score agreement between the

instructor and each of the inspectors at every supply for both answer type (Yes/No and risk level), pre- and post-training.

### 2.5.2. Coefficient of Variation

Coefficient of variation analysis was used to determine SI score variability between the inspectors at every supply for both answer type (Yes/No and risk level), pre- and post-training.

### 2.5.3. Index of Dispersion

Index of dispersion analysis was used to determine the consistency of answer agreement between the inspectors for each individual SI question for both answer types (Yes/No and risk level), pre- and post-training. An example of this may be seen in Table 2. The indexes of dispersion for each individual question were further processed in two different ways to calculate (a) the mean index of dispersion per individual question (1–12), pre- and post-training, and (b) the mean index of dispersion per individual supply (1–36), pre- and post-training.

**Table 2.** Example of answer index of dispersion values for each individual question from supply 1.

| Supply 1 | Q1 | Q2 | Q3 | Q4 | Q5 | Q6 | Q7 | Q8 | Q9 | Q10 | Q11 | Q12 |
|---|---|---|---|---|---|---|---|---|---|---|---|---|
| High | 1 | 3 | 2 | 1 | 1 | 3 | 2 | 3 | 0 | 1 | 0 | 0 |
| Medium | 3 | 1 | 1 | 2 | 1 | 0 | 1 | 0 | 0 | 0 | 3 | 0 |
| Low | 0 | 0 | 0 | 0 | 0 | 0 | 0 | 0 | 0 | 1 | 1 | 0 |
| No | 0 | 0 | 1 | 1 | 2 | 1 | 1 | 1 | 4 | 2 | 0 | 4 |
| Sum squares | 10 | 10 | 6 | 6 | 6 | 10 | 6 | 10 | 16 | 6 | 10 | 16 |
| Index of dispersion | 0.500 | 0.500 | 0.833 | 0.833 | 0.833 | 0.500 | 0.833 | 0.500 | 0.000 | 0.833 | 0.500 | 0.000 |

(a)    The mean index of dispersion per individual question (1–12)

Taking question 1 as an example, the indexes of dispersion from every supply were used to determine the mean index of dispersion for question 1 answers, pre- and post-training. This was repeated for questions 2–12 to determine which questions showed the most inconsistency in answer agreement between the inspectors.

(b)    The mean index of dispersion per individual supply (1–36)

Taking supply 1 as an example, the indexes of dispersion of the answers to questions 1–12 at supply 1 were used to determine the mean index of dispersion at that supply, pre- and post-training. This was repeated for supplies 2–36 to determine which supplies exhibited the highest mean index of dispersion due to inconsistent answer agreement.

Microsoft Excel was used to undertake all statistical analysis.

### 2.6. Missing Data Treatment

On 37 random occasions, an identified risk was accidentally not assigned a risk level by one of the inspectors. The research team chose to apply a "High" risk to these answers as we consider the identification of a risk without further context to warrant a high risk rating. To analyse the sensitivity of this approach, a "Low" and "Medium" risk was applied to each of the 37 questions to determine how this would affect the statistical outcomes.

### 2.7. Research Ethics

The University of Surrey Self-Assessment For Ethics (SAFE) tool was used to confirm this project met all ethical requirements as outlined by the University of Surrey Research Ethics Committee.

## 3. Results

*3.1. SI Score Correlation and Agreement between the Instructor and Each Inspector, Pre- and Post-Training*

Training resulted in improved SI score correlation for Yes/No questions and risk level questions between the instructor and each of the inspectors (Table 3); improved SI score agreement (Yes/No) between the instructor and each of the inspectors (Table 4); and improved SI score agreement (risk level) between the instructor and inspectors 1 and 3 (Table 4). However, training led to less agreement (risk level) between the instructor and inspectors 2 and 4 (Table 4).

**Table 3.** SI score (Yes/No and risk level) correlation (Pearson's r) between the instructor and each inspector, pre- and post-training.

| Yes/No SI Scores | | Inspector 1 | Inspector 2 | Inspector 3 | Inspector 4 |
|---|---|---|---|---|---|
| Pre-training | Pearson's r | (34) = 0.313, $p = 0.063$ | (34) = 0.216, $p = 0.206$ | (34) = 0.465, $p = 0.004$ | (34) = 0.374, $p = 0.025$ |
| Post-training | Pearson's r | (34) = 0.686, $p < 0.001$ | (34) = 0.399, $p = 0.016$ | (34) = 0.531, $p = 0.001$ | (34) = 0.533, $p = 0.001$ |
| **Risk level SI scores** | | | | | |
| Pre-training | Pearson's r | (34) = 0.351, $p = 0.036$ | (34) = 0.308, $p = 0.068$ | (34) = 0.292, $p = 0.084$ | (34) = 0.396, $p = 0.017$ |
| Post-training | Pearson's r | (34) = 0.748, $p < 0.001$ | (34) = 0.409, $p = 0.013$ | (34) = 0.606, $p < 0.001$ | (34) = 0.572, $p < 0.001$ |

**Table 4.** SI score (Yes/No and risk level) agreement (Lin's concordance correlation coefficient (CCC)) between the instructor and each inspector, pre- and post-training.

| Yes/No SI Scores | | Inspector 1 | Inspector 2 | Inspector 3 | Inspector 4 |
|---|---|---|---|---|---|
| Pre-training | Lin's CCC (95% confidence intervals) | 0.231 (−0.02–0.45) | 0.122 (−0.07–0.30) | 0.332 (0.10–0.53) | 0.361 (0.05–0.61) |
| Post-training | Lin's CCC (95% confidence intervals) | 0.680 (0.46–0.82) | 0.353 (0.07–0.58) | 0.408 (0.17–0.60) | 0.517 (0.24–0.72) |
| **Risk level SI scores** | | | | | |
| Pre-training | Lin's CCC (95% confidence intervals) | 0.318 (0.03–0.56) | 0.277 (−0.02–0.53) | 0.287 (−0.04–0.56) | 0.340 (0.06–0.57) |
| Post-training | Lin's CCC (95% confidence intervals) | 0.607 (0.41–0.75) | 0.201 (0.03–0.36) | 0.405 (0.20–0.58) | 0.308 (0.13–0.47) |

*3.2. SI Score Variability between Inspectors, Pre- and Post-Training*

Although training had an overall positive impact on reducing SI score variability (Yes/No and risk level) between inspectors, variability of SI scores remained at certain supplies (Table 5; Figures 2 and 3). Such variability is a result of inconsistent answering of individual SI questions by inspectors. Taking supply 31 in Figure 2 as an example, there is high SI score variability even post-training due to each of the Yes/No questions 1, 3, 4, 5, 12 exhibiting a 50:50 divide in answer response type between the four inspectors.

**Table 5.** Variability (coefficient of variation) of Yes/No and risk level SI scores between all inspectors across all supplies (1–36), pre- and post-training.

| Yes/No SI Scores | Mean SI Score Coefficient of Variation across 36 Supplies | Standard Deviation |
| --- | --- | --- |
| Pre-training | 21.25 | 8.13 |
| Post-training | 16.16 | 6.87 |
| **Risk level SI scores** | | |
| Pre-training | 24.12 | 10.29 |
| Post-training | 19.62 | 10.74 |

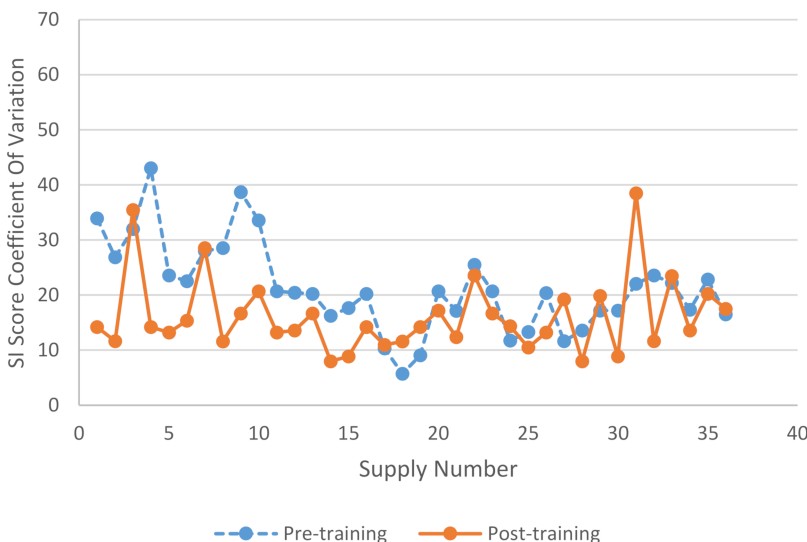

**Figure 2.** Variability (coefficient of variation) of Yes/No SI scores between all inspectors per supply (1–36), pre- and post-training.

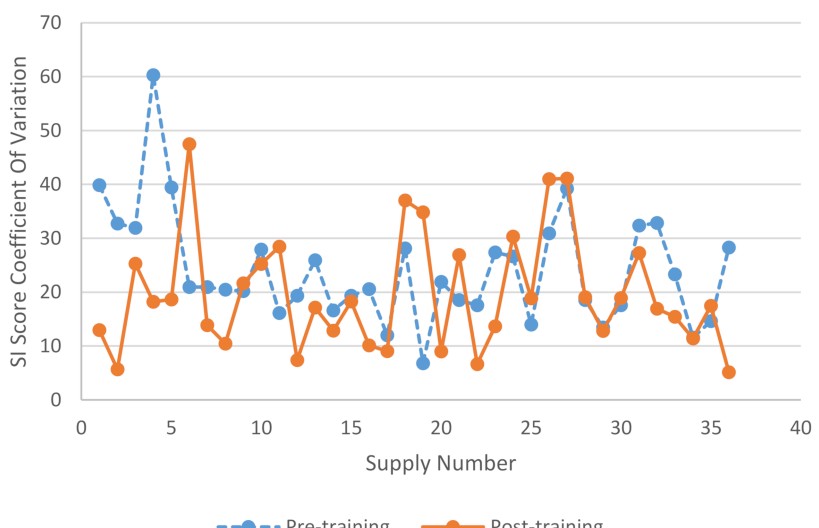

**Figure 3.** Variability (coefficient of variation) of risk level SI scores between all inspectors per supply (1–36), pre- and post-training.

*3.3. Consistency of Answer Agreement between Inspectors for Each Individual SI Question, Pre- and Post-Training*

　　Answer agreement between inspectors was determined for each individual SI question at every supply, pre- and post-training. Answer agreement for each question, as measured by index of dispersion, was processed to determine: (a) the mean answer agreement for each individual question (1–12), pre- and post-training, and (b) the mean answer agreement for each supply (1–36), pre- and post-training.

(a)　Training improved mean answer (Yes/No and risk level) agreement of SI questions (1–12) (Table 6), though comparable standard deviations pre- and post-training for both answer types indicate certain questions still exhibited nonagreement. Answer agreement for Yes/No questions improved post-training for questions 2–5 and 8–11. Slight decreases in answer agreement were observed for questions 1, 6, 7 and 12 (Figure 4). When estimating risk level, training improved answer agreement for questions 1–5 and 8–11. Slight decreases in answer agreement were observed for questions 6, 7 and 12 (Figure 5).

(b)　Training improved mean answer (Yes/No and risk level) agreement per supply (Table 7), however, increases in standard deviation values post-training for both answer types indicate sustained/increased nonagreement of answers at some supplies, for example, supply 31. Greater answer nonagreement was observed in estimation of risk level as opposed to Yes/No questions. Of the seven supplies (6,9,24,27,28,31,36) that exhibited lower mean Yes/No answer agreement post-training, four of these supplies (6,9,24,31) also exhibited lower mean risk level answer agreement post-training (Figures 6 and 7).

**Table 6.** Answer agreement (index of dispersion) between inspectors across all Yes/No and risk level SI questions, pre- and post-training.

| Yes/No SI Scores | Mean Answer Index of Dispersion across 12 Questions | Standard Deviation |
|:---:|:---:|:---:|
| Pre-training | 0.41 | 0.28 |
| Post-training | 0.27 | 0.21 |
| **Risk level SI scores** | | |
| Pre-training | 0.55 | 0.12 |
| Post-training | 0.41 | 0.21 |

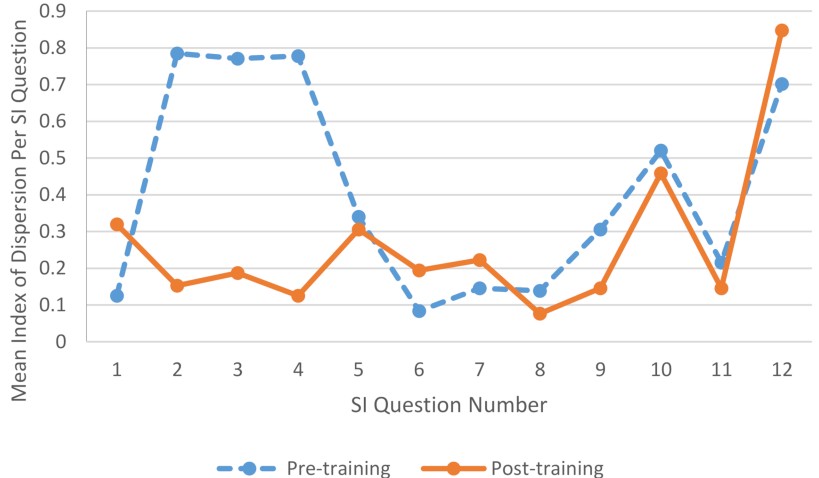

**Figure 4.** Answer agreement (index of dispersion) between inspectors per Yes/No SI question (1–12), pre- and post-training.

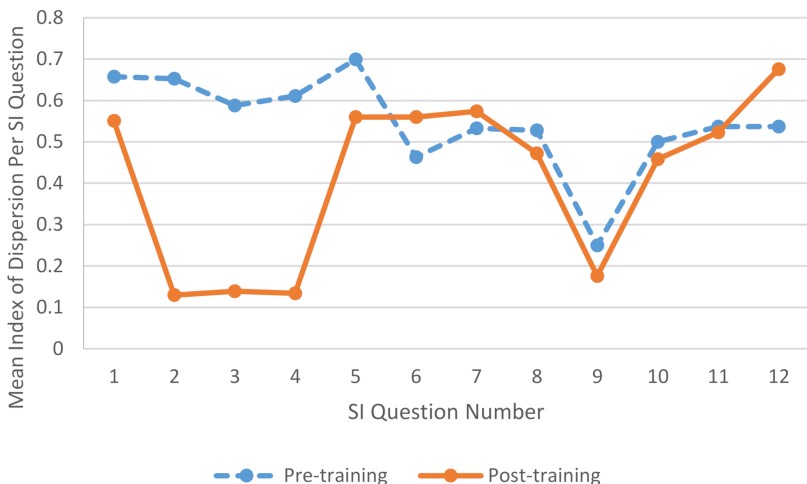

**Figure 5.** Answer agreement (index of dispersion) between inspectors per risk level SI question (1–12), pre- and post-training.

**Table 7.** Yes/No and risk level answer agreement (index of dispersion) between inspectors across 36 supplies, pre- and post-training.

| Yes/No SI Scores | Mean Answer Index of Dispersion across 36 Supplies | Standard Deviation |
|---|---|---|
| Pre-training | 0.41 | 0.10 |
| Post-training | 0.27 | 0.14 |
| **Risk level SI scores** | | |
| Pre-training | 0.55 | 0.08 |
| Post-training | 0.41 | 0.12 |

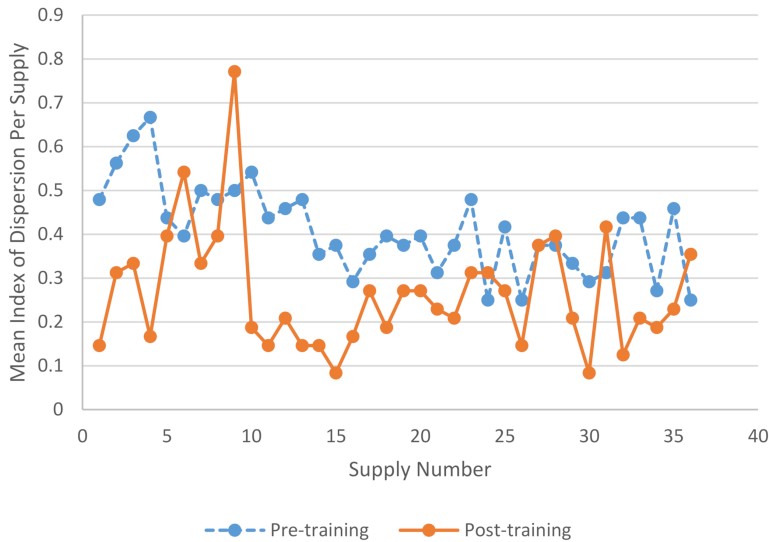

**Figure 6.** Yes/No answer agreement (index of dispersion) between inspectors per supply (1–36), pre- and post-training.

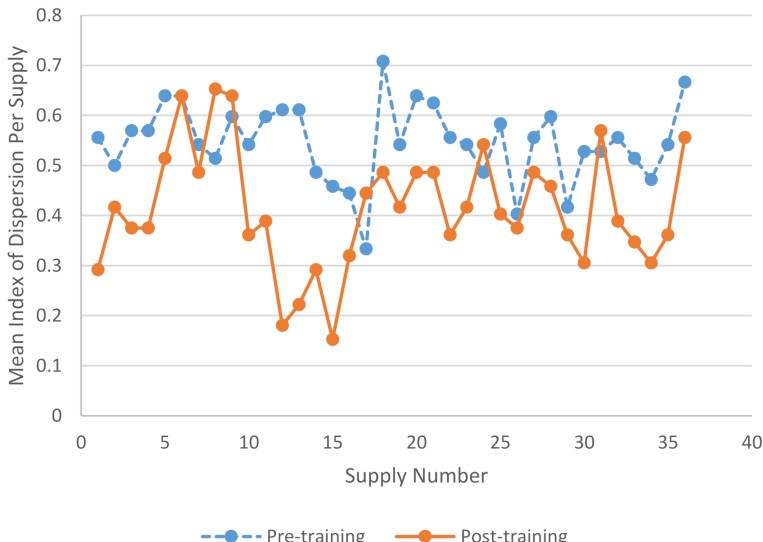

**Figure 7.** Risk level answer agreement (index of dispersion) between inspectors per supply (1–36), pre- and post-training.

### 3.4. Missing Data Treatment

Where "Low" and "Medium" were applied instead of a "High" to each of the 37 unanswered risk level questions to determine how this would affect the statistical outcomes, no changes of significance were observed. This is attributed to the spread of missing data across inspectors, supplies and studies; the low percentage of missed risk level answers (1.07%); and the use of averaging during statistical analysis.

## 4. Discussion

The strengths of SIs are their simplicity, flexibility and adaptability [6]. One of the challenges associated with SIs identified by Pond et al. [6] was the potential for inconsistent agreement of answers between inspectors. This may be caused by overly technical SI questions or by different perceptions of risk due to varied experiences and levels of inspector expertise. Training for inspectors is paramount prior to SI use [6]. Though there are studies from other sectors that examine the effects of training on inter-observer (inspector) answer agreement [36–39], to our knowledge, this is the first to gauge the effect of training on SI inter-inspector answer agreement. Okotto-Okotto et al. and Yentumi et al. [29,30] studied levels of inter-inspector SI score agreement, however, their study designs meant participating inspectors had varying levels of technical knowledge and inspections were undertaken at different times under varying conditions for each inspector. Okotto-Okotto et al. [29] provided standardised training to inspectors, though this was prior to study commencement, meaning there was no pre-training baseline data to compare against.

To gauge the effects of training on consistency of answer agreement between inspectors, every aspect of study 1 and 2's design was kept constant as far as reasonably possible, including inspector age bracket, education and experience; supply locations; and weather. Okotto-Okotto et al. [29] undertook a similar study in which inspectors carried out inspections at each supply independently to avoid influencing of results. We argue that the dynamic and often transient nature of risk meant all inspectors should inspect each supply at the same time. This is especially relevant in Uganda where rapid rainfall/evaporation cycles occur [40], exposing risk that may not be obvious otherwise. We partially emulated the Okotto-Okotto et al. [29] study design by gauging SI score correlation and agreement between an instructor and inspectors, pre- and post-training. There were increases in correlation between the SI scores of the instructor and each of the inspectors for both answer types (Yes/No and risk level) post-training (Table 3). Pearson's correlation coefficients with $p$-values of <0.02 calculated

for all SI score correlations between the instructor and each of the inspectors post-training indicated a strong significance of the correlation.

Correlation analysis determines the strength of the linear relationship between two variables, but not the level of agreement. None of the inspectors exhibited complete SI score agreement with the instructor, however, Lin's CCC values confirmed there was increased SI score agreement between the instructor and each of the inspectors post-training for the Yes/No SI answer type (Table 4). For the risk level SI scores, only inspectors 1 and 3 showed improved SI score agreement with the instructor post-training. This highlights the difference between correlation and agreement analysis as, despite the increase in risk level SI score correlation between the instructor and each of the inspectors, SI score agreement between the instructor and inspectors 2 and 4 exhibited a slight decrease post-training. This may be attributed to the greater requirement of the inspector to apply subjective thinking to the risk level method of answering.

Where it is not practical to gauge SI score agreement against an instructor or experienced inspector, for example, in such cases where all inspectors are experienced and trained to a high level, it is still beneficial to understand the consistency of answer agreement within an inspector group. This allows for confidence in comparing SI scores from different inspectors within a surveillance or management programme. Our results show inconsistency in interpretation of risk by inspectors, with improved consistency following the provision of training. One limitation of the SI score method is that a lone metric does not provide context regarding how the final score was derived, i.e., which risk factors were identified. This limitation is more evident for the risk level method of scoring, e.g., an SI score of 10 could be a result of ten low (1) risks or two high (5) risks. As expected, due to the higher scoring scale (0–60) in the risk level SI score method, variability of SI scores between inspectors was greater than for the Yes/No (scale 1–12) scoring method post-training (Table 5). To provide context to the improved SI score agreement between inspectors, and to prove this was not due to random occurrence, the consistency of answer agreement between inspectors for each individual SI question was investigated using index of dispersion analysis (Table 6). The results from this analysis confirm that the improved agreement of SI scores post-training is not random but rather due to improved answer agreement. As with the SI score agreement analysis, due to the higher scoring scale (0–60) and increased number of potential answer options in the risk level method (e.g., No, Low, Medium, High), inconsistency/dispersion of risk level answers was higher than for the Yes/No method post-training.

Pond et al. [6] discussed the challenges related to interpretation of SI questions by inspectors. Such challenges were deemed especially prevalent where SI questions are not specific to the local context or where they have been adapted without thorough consultation with supply stakeholders and rigorous field pilot testing. Inaccuracy of SI question structure, or question phrasing that may lead to variations in interpretation among inspectors, especially without the provision of additional guidance or operational definitions, was also noted as a primary challenge towards achieving consistency of answer agreement between inspectors. Linguistic challenges were also identified. These refer not only to interpretation and translation of questions between languages, but also to the difficulty in structuring an SI question so that a standard answer type (e.g., all "Yes" answers) will always indicate a risk. The combination of SI question and accompanying explanatory note used in this study was designed to remove any challenges related to question clarity. There were, however, supply component names (e.g., spring box) that laypersons, or experienced inspectors with limited knowledge of spring supplies, may have been unfamiliar with.

Training improved consistency of answer agreement between inspectors for the majority of the 12 individual SI questions. Questions 6 and 7, which query the "adequacy" of fencing around the spring supply, and question 12, which focusses on points of entry to the aquifer within 100 metres of the supply, exhibited slightly less agreement post-training for both answer types (Yes/No and risk level). Question 1, which references the spring box component, also exhibited slightly less agreement post-training for the Yes/No answer type only. In such cases where inconsistency of answer agreement between inspectors remains an issue for specific questions post-training, targeted question-specific training

and/or rephrasing of questions may be required to rectify uncertainties. Four supplies (6,9,24,31) exhibited less agreement between inspectors post-training for both answer types (Figures 6 and 7). This infers that those particular supplies may have design features or contexts that are more difficult to assess in terms or risk and therefore increase the inconsistency of risk perception among inspectors.

Despite the increasing significance of SIs as a crucial component of water safety management, the frequency with which risk assessments are undertaken at many supplies may still be irregular and infrequent [31]. The impacts of inconsistent or inaccurate SI scores are numerous, e.g., incorrect prioritisation of remedial works or continued use of a contaminated supply. Tailored training and additional guidance for carrying out SIs consistently and accurately is key to ensuring SIs are effective in achieving/maintaining safe drinking water. Though neither water quality testing nor SIs can be used alone to define the safety of a water supply, both activities should holistically contribute towards a current and historic safety profile of a supply. In countries such as Uganda that are still experiencing prevalent waterborne diseases [41], limited resources for water treatment and testing places even greater importance on the need for accurate and precise SI results. Buy-in and continued support for SIs from communities, stakeholders and policy makers are key to their success.

The opportunity to compare SI results both within and between regions is one of the main advantages of the activity. To achieve accurate comparisons, standardisation of SI forms is vital. However, an SI form should not be a static document but rather one that may be updated where questions are proven to be scientifically invalid or where they need to be rephrased due to increased answer inconsistency. The tailoring of SI forms to local contexts is promoted by the WHO [34] and should be practiced where practical. The tailoring of questions to the Ugandan context may have improved answer agreement between inspectors in this study. Where SI results are to be used to prioritise valuable supply protection interventions (e.g., Cronin et al. [42]), the importance of SI score inter-inspector agreement is paramount. Suitable training of inspectors relative to their knowledge and experience is required to avoid imprecision and inaccuracy of results. Where "gold standard" inspectors are used to train others, it is vital that they themselves are trained to the highest possible level.

The limitations of this study include the sample size and only one water supply type being examined during the study. Sample size was limited due to availability of suitable inspectors and the accessibility of appropriate spring supplies. If similar studies to this one are undertaken, we recommend inspecting different supply types (such as dug wells) and the inclusion of an additional semi-structured interview with inspectors subsequent to completing all inspections to obtain feedback on any particular issues regarding individual SI questions. Although not considered essential for this study, the use of a control group with no formal training inputs would be interesting to investigate experiential learning and familiarisation separately from directed learning.

## 5. Conclusions

Though there have been limited studies on SI inter-inspector answer agreement [29,30], ours was the first to make use of a systematic before/after comparison to specifically determine the effects of training on answer agreement. Consistent identification of risk by inspectors is vital to ensure the safety of drinking-water supplies. The results from our study confirm that more in-depth training is essential in such cases where perceptions play a part in answering a question, e.g., as part of risk identification. This validates recommendations by Pond et al. and the WHO [6,28] that state the importance of SI inspector training prior to use. The observed increases in the consistency of answer agreement per SI question and improvements in SI score agreement between inspectors post-training suggests a heuristic approach to undertaking SIs is not suitable for achieving the most accurate or consistent results within an inspector group. The knowledge base and experience level of inspectors should be determined before a suitable training schedule is developed based on their abilities. Such additional training will add a degree of objectivity to inspector observations. Analysis of answer agreement for each individual SI question pre- and post-training confirmed (a) increased SI score agreement between inspectors was not random but was due to improved answer agreement per SI question and (b) SI

questions typically exhibited greater answer agreement post-training with the exception of certain questions that require rephrasing and/or additional training. The results from this study were not dictated by specific research design or location characteristics, therefore the method may be referenced by future similar studies. The study findings may be used to advocate for appropriate SI training within organisations.

**Author Contributions:** Conceptualisation, R.K.; methodology, R.K.; formal analysis, R.K.; writing—original draft preparation, R.K.; writing—review and editing, J.H., K.O., D.J.L., R.M.M., R.M. and K.P. All authors have read and agreed to the published version of the manuscript.

**Funding:** This research was funded by Research England as part of the government's aid strategy to further the sustainable development and welfare of developing countries.

**Conflicts of Interest:** Rory Moses McKeown is a WHO consultant. The authors alone are responsible for the views expressed in this publication and they do not necessarily represent the decision or stated policy of the World Health Organization.

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
