# Peer review of "Does Training Improve Sanitary Inspection Answer Agreement between Inspectors? Quantitative Evidence from the Mukono District, Uganda"

_resources, doi:10.3390/resources9100120_

Round 1

Reviewer 1 Report

This article contains very important information and answers the journal's theme.

However, I advise the author to rectify the comments that I presented throughout the article, most of these comments relate to minor corrections, then to update them.

Reviewer 2 Report

The manuscript concerns the important issue of the investigation the effect of training on inspector risk perception and inter-inspector answer agreement. Modified SI forms were developed and applied in the study, where SI questions were designed using both the traditional ‘Yes/No’ answer type and the risk level assignment answer type, that is commonly utilised in WSPs. My comments are as follows: What is the definition of the risk? In what way the presented analysis will help to achieve the safety of drinking-water supplies? Are there concrete steps that can be recommended and how generalizable are the findings? Can they be applied to other areas? How dependent are they to specific characteristics of the region under examination. How in practice the results of the presented analysis can be used. The conclusion should be organized in such a way that it shows a summary of Authors main achievements during the research You have done.
